# A neural correlate of individual odor preference in *Drosophila*

**Matthew A Churgin[1,2†], Danylo O Lavrentovich[1,2†], Matthew A-Y Smith[1,2‡], Ruixuan Gao[3,4,5§], Edward S Boyden[3,6,7,8,9], Benjamin L de Bivort[1,2*]**

[1]Organismic and Evolutionary Biology, Harvard University, Cambridge, United States; [2]Center for Brain Science, Harvard University, Cambridge, Cambridge, United States; [3]McGovern Institute, MIT, Cambridge, United States; [4]MIT Media Lab, MIT, Cambridge, United States; [5]Janelia Research Campus, Howard Hughes Medical Institute, Ashburn, United States; [6]Department of Biological Engineering, MIT, Cambridge, United States; [7]Koch Institute, Department of Biology, MIT, Cambridge, United States; [8]Howard Hughes Medical Institute, Chevy Chase, United States; [9]Department of Brain and Cognitive Sciences, MIT, Cambridge, United States

**\*For correspondence:**
debivort@oeb.harvard.edu

†These authors contributed equally to this work

**Present address:** ‡Department of Biology, Illinois Institute of Technology, Chicago, United States; §Department of Biological Sciences/Chemistry, University of Illinois Chicago, Chicago, United States

## eLife Assessment

What makes one member of the species behave differently from another? This is a core problem in behavioral neuroscience. This **valuable** study seeks an answer for the specific case of the fruit fly expressing preferences for one odor over another. By a combination of behavioral measurements, neurophysiology, and network modeling, the authors find **solid** evidence for at least one locus of individuality in the peripheral olfactory system.

**Abstract** Behavior varies even among genetically identical animals raised in the same environment. However, little is known about the circuit or anatomical origins of this individuality. Here, we demonstrate a neural correlate of *Drosophila* odor preference behavior in the olfactory sensory periphery. Namely, idiosyncratic calcium responses in projection neuron (PN) dendrites and densities of the presynaptic protein Bruchpilot in olfactory receptor neuron (ORN) axon terminals correlate with individual preferences in a choice between two aversive odorants. The ORN-PN synapse appears to be a locus of individuality where microscale variation gives rise to idiosyncratic behavior. Simulating microscale stochasticity in ORN-PN synapses of a 3062 neuron model of the antennal lobe recapitulates patterns of variation in PN calcium responses matching experiments. Conversely, stochasticity in other compartments of this circuit does not recapitulate those patterns. Our results demonstrate how physiological and microscale structural circuit variations can give rise to individual behavior, even when genetics and environment are held constant.

## Introduction

Individuality is a fundamental aspect of behavior that is observed even among genetically identical animals reared in similar environments. We are specifically interested in individuality that is evident as idiosyncratic differences in behavior that persist for much of an animal's lifespan. Such variability is observed across species, including round worms (*Stern et al., 2017*), aphids (*Schuett et al., 2011*), fish (*Laskowski et al., 2022*), mice (*Freund et al., 2013*), and people (*Johnson et al., 2010*). Small, genetically tractable model species, such as *Drosophila*, are particularly promising for discovering the genetic and neural circuit basis of individual behavior variation. Flies exhibit individuality in many behaviors (*Werkhoven et al., 2021*), and the mechanistic origins of this variation have been studied

for phototactic preference (*Kain et al., 2012*), temperature preference (*Kain et al., 2015*), locomotor handedness (*Ayroles et al., 2015*; *Buchanan et al., 2015*; *de Bivort et al., 2022*), object-fixated walking (*Linneweber et al., 2020*), and odor preference (*Honegger et al., 2020*). Generally, the neural substrates of individuality are poorly understood, though in a small number of instances nanoscale circuit correlates of individual behavioral biases have been identified (*Lillvis et al., 2022*; *Linneweber et al., 2020*; *Skutt-Kakaria et al., 2019*). We hypothesize that as sensory cues are encoded and transformed to produce motor outputs, their representation in the nervous system becomes increasingly idiosyncratic and predictive of individual behavioral responses. An alternative hypothesis is that neural representations are the same across individuals and individuality emerges through biomechanical differences and interactions with the environment. We seek to determine if 'loci of individuality' – sites at which this idiosyncrasy emerges – exist, and if so, where in the sensorimotor cascade.

Olfaction in the fruit fly *Drosophila melanogaster* is an amenable sensory system for identifying loci of individuality as (1) individual odor preferences can be recorded readily, (2) neural representations of odors can be measured via calcium imaging, (3) the circuit elements of the pathway are well-established, and (4) a deep genetic toolkit enables mechanism-probing experiments. The neuroanatomy of the olfactory system, from the antenna through its first central-brain processing neuropil, the antennal lobe (AL), is broadly stereotyped across individuals (*Couto et al., 2005*; *Grabe et al., 2015*; *Wilson et al., 2004*). The AL features ~50 anatomically identifiable microcircuits called glomeruli (*Figure 1A*). Each glomerulus represents an odor-coding channel and receives axon inputs from olfactory receptor neurons (ORNs) expressing the same olfactory receptor gene (*de Bruyne et al., 2001*). Uniglomerular projection neurons (PNs) carry odor information from each glomerulus deeper into the brain (*Jeanne and Wilson, 2015*). AL-intrinsic local neurons (LNs) project among glomeruli (*Chou et al., 2010*) and modulate odor representations (*Wilson and Laurent, 2005*). Glomerular organization is a key stereotype of the AL; using glomeruli as landmarks, one can identify comparable ORN axons and PNs across individuals.

Individual flies differ in their PN calcium responses to identical odor stimuli, as well as their odor-vs.-odor preference choices (*Honegger et al., 2020*). Several possible determinants of individual odor preference can already be hypothesized for the fly olfactory circuit (*Rihani and Sachse, 2022*). The extent of preference variability depends on dopamine and serotonergic modulation (*Honegger et al., 2020*). Neuromodulation clearly plays a role in the regulation of behavioral individuality (*Maloney, 2021*), but its effects vary by modulator and behavior (*de Bivort et al., 2022*; *Kain et al., 2012*). With respect to wiring variation, the number of ORNs and PNs innervating a given glomerulus varies within hemispheres (*Tobin et al., 2017*) and across individuals (*Grabe et al., 2016*; *Schlegel et al., 2020*), as does the glomerulus-innervation pattern of individual LNs (*Chou et al., 2010*). Subpopulations of LNs and PNs express variable serotonin receptors (*Sizemore and Dacks, 2016*), so the effects of neuromodulation and wiring may interact to influence individuality. Little is known about possible molecular or nanoscale correlates of individual behavioral bias. Thus, individual odor preference could have its origins in many potential mechanisms, ranging from circuit wiring to modulation to neuronal intrinsic properties.

Outside the olfactory system, there are a few examples in which microscale circuit variation predicts individual behavioral preference. Wiring asymmetry in an individual fly's dorsal cluster neurons is predictive of the straightness of its object-oriented walking behavior (*Linneweber et al., 2020*), and left-right asymmetry in the density of presynaptic sites of protocerebral bridge to lateral accessory lobe-projecting neurons predicts an individual fly's idiosyncratic turning bias (*Skutt-Kakaria et al., 2019*). The number of synaptic connections from the pC2l to pIP10 neurons correlates with male song rate during courtship (*Lillvis et al., 2022*), and the presence of ectopic branches in neurons of the T2 hemilineage predicts delayed spontaneous flight initiation (*Mellert et al., 2016*).

In this work, we sought to identify loci of individuality by measuring odor preferences and neural responses to odors in the same individuals and determining the extent to which the latter predicted the former. We found that idiosyncratic calcium responses in PNs were correlated with individual preferences in a choice between two aversive odorants. Examining a molecular component presynaptic to PNs, we found that the density of the scaffolding protein Bruchpilot also predicts odor preference. To unify these results and connect wiring variation to circuit outputs and behavior, we simulated developmental variation in a 3062-neuron spiking model of the AL. Simulated stochasticity in the ORN-PN synapse recapitulated our empirical findings. Thus, we identified the ORN-PN synapse as a likely locus

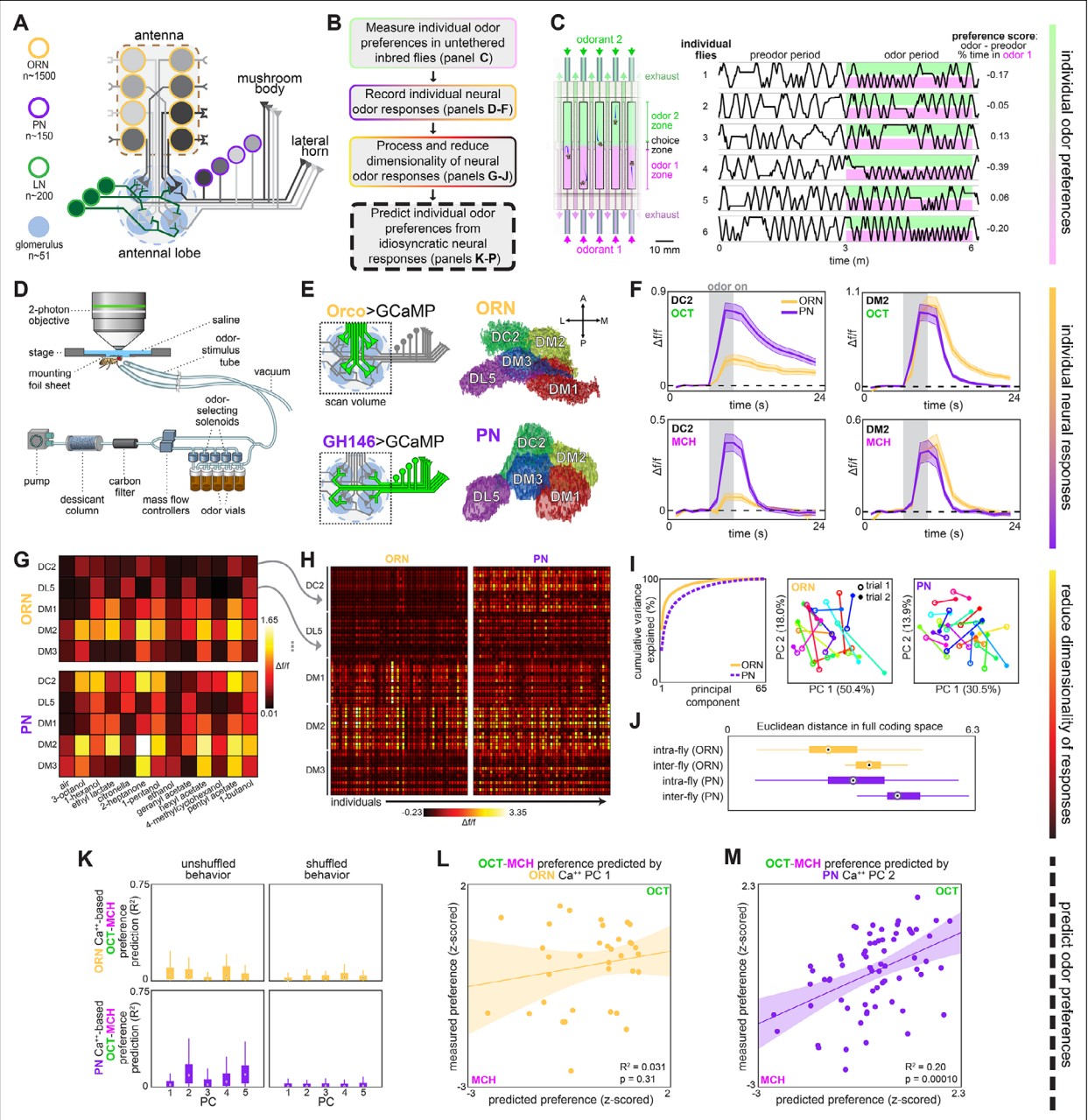

**Figure 1.** Idiosyncratic calcium dynamics predict individual odor preferences. (**A**) Olfactory circuit schematic. Olfactory receptor neurons (ORNs, peach outline) and projection neurons (PNs, plum outline) are comprised of ~51 classes corresponding to odor receptor response channels. ORNs (gray shading) sense odors in the antennae and synapse on dendrites of PNs of the same class in ball-shaped structures called glomeruli located in the antennal lobe (AL). Local neurons (LNs, green outline) mediate interglomerular cross-talk and presynaptic inhibition, amongst other roles (*Olsen and Wilson, 2008*; *Yaksi and Wilson, 2010*). Odor signals are normalized and whitened in the AL before being sent to the mushroom body and lateral horn for further processing. Schematic adapted from Figure 2C of *Honegger et al., 2020*. (**B**) Experiment outline. (**C**) Odor preference behavior tracking setup (reproduced from Figure 1B of *Honegger et al., 2020*) and example individual fly ethograms. OCT (green) and MCH (magenta) were presented for 3 minutes. (**D**) Head-fixed two-photon calcium imaging and odor delivery setup (reproduced from Figure 2A of *Honegger et al., 2020*). (**E**) Orco and GH146 driver expression profiles (left) and example segmentation masks (right) extracted from two-photon calcium images for a single fly expressing Orco>GCaMP6m (top, expressed in a subset of all ORN classes) or GH146>GCaMP6m (bottom, expressed in a subset of all PN classes). (**F**) Time-dependent Δf/f for glomerular odor responses in ORNs (peach) and PNs (plum) averaged across all individuals: DC2 to OCT (upper left), DM2 to OCT (upper right), DC2 to MCH (lower left), and DM2 to OCT (lower right). Shaded error bars represent S.E.M. (**G**) Peak Δf/f for each glomerulus-odor pair averaged across all flies. (**H**) Individual neural responses measured in ORNs (left) or PNs (right) for 50 flies each. Columns represent the average of up to four odor responses from a single fly. Each row represents one glomerulus-odor response pair. Odors are the same as in panel (**G**). (**I**) Principal component analysis of individual neural responses. Fraction of variance explained vs. principal component number (left). Trial 1 and trial 2

*Figure 1 continued*

of ORN (middle) and PN (right) responses for 20 individuals (unique colors) embedded in PC 1–2 space. (**J**) Euclidean distances between glomerulus-odor responses within and across flies measured in ORNs (n = 65 flies) and PNs (n = 122 flies). Distances calculated without PCA compression. (**K**) Bootstrapped $R^2$ of OCT-MCH preference prediction from each of the first five principal components of neural activity measured in ORNs (top, all data) or PNs (bottom, training set). (**L**) Measured OCT-MCH preference vs. preference predicted from PC 1 of ORN activity (n = 35 flies). (**M**) Measured OCT-MCH preference vs. preference predicted from PC 2 of PN activity in n = 69 flies using a model trained on a training set of n = 47 flies (see *Figure 2—figure supplement 1C and D* for train/test flies analyzed separately). Shaded regions in (**L, M**) are the 95% CIs of the fit estimated by bootstrapping. In (**J, K**), points represent the median value, boxes represent the interquartile range, and whiskers the range of the data.

The online version of this article includes the following figure supplement(s) for figure 1:

**Figure supplement 1.** Behavioral measurements and individual preference persistence.

**Figure supplement 2.** Average glomerulus-odor time-dependent responses.

**Figure supplement 3.** Individual glomerulus-odor responses.

**Figure supplement 4.** Correspondence in calcium responses between lobes and trials.

**Figure supplement 5.** Glomerulus responses and identification.

**Figure supplement 6.** Idiosyncrasy of olfactory receptor neuron (ORN) and projection neuron (PN) responses.

**Figure supplement 7.** Calcium response correlation matrices.

**Figure supplement 8.** Calcium imaging principal component loadings.

**Figure supplement 9.** Estimating latent calcium–behavior correlations.

**Figure supplement 10.** OCT-AIR preference prediction.

of individuality in fly odor preference, demonstrating that behaviorally-relevant variation in neural circuits can be found in the sensory periphery at the nanoscale.

## Results

### Individual flies encode odors idiosyncratically

Focusing on behavioral variation within a genotype, we used isogenic animals expressing the fluorescent calcium reporter GCaMP6m (*Chen et al., 2013*) in either of the two most peripheral neural subpopulations of the *Drosophila* olfactory circuit, ORNs or PNs (*Figure 1E*). We performed head-fixed two-photon calcium imaging after measuring odor preference in an untethered assay (*Honegger et al., 2020*; *Figure 1B–D*, *Figure 1—figure supplement 1A*; *Videos 1 and 2*). Individual odor preferences are stable over timescales longer than this experiment (*Figure 1—figure supplement 1B–E*).

We measured volumetric calcium responses in the AL, where ORNs synapse onto PNs in ~50 discrete microcircuits called glomeruli (*Figure 1A*; *Couto et al., 2005*; *Grabe et al., 2015*). Flies were stimulated with a panel of 12 odors plus air (*Figure 1D*, *Figure 1—figure supplement 2*) and *k*-means clustering was used to automatically segment the voxels of five glomeruli from the resulting 4-D calcium image stacks (*Figure 1E*, *Figure 1—figure supplement 5*, 'Materials and methods'; *Couto et al., 2005*). Both ORN and PN odor responses were roughly stereotyped across individuals (*Figure 1G and H*), but also idiosyncratic (*Honegger et al., 2020*). Responses in PNs appeared to be more idiosyncratic than ORNs (*Figure 1J*); a logistic

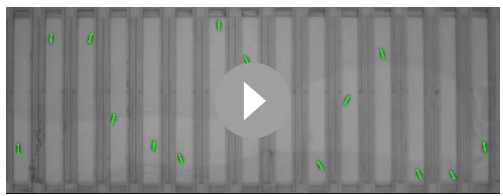

**Video 1.** Example recording with automated tracking of an odor-vs.-air behavioral assay. The recent positions of each fly (green line) are shown in different colors. Red bar indicates when the odor stream is turned on.
https://elifesciences.org/articles/90511/figures#video1

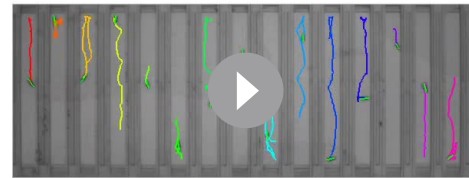

**Video 2.** Example recording with automated tracking of an odor-vs.-odor behavioral assay. The recent positions of each fly (green line) are shown in different colors. Magenta and green bars at right indicate when MCH and OCT are respectively flowing into the top and bottom halves of each arena.
https://elifesciences.org/articles/90511/figures#video2

linear classifier decoding fly identity from glomerular responses was more accurate when trained on PN than ORN responses (*Figure 1—figure supplement 6A*). While the responses of single ORNs are known to vary more than those of single PNs (*Wilson, 2013*), our recordings capture the total response of all ORNs or PNs in a glomerulus. This might explain our observation that ORNs exhibited less idiosyncrasy than PNs. PN responses were more variable within flies, as measured across the left and right hemisphere ALs, compared to ORN responses (*Figure 1—figure supplement 6C*; p<2 × 10⁻⁵, Mann–Whitney *U* test), suggesting that odor representations become more divergent farther from the sensory periphery.

## PN, but not ORN, responses predict odor-vs.-odor preference

Next we analyzed the relationship of idiosyncratic coding to odor preference, by asking in which neurons (if any) did calcium responses predict individual preferences of flies choosing between two aversive monomolecular odors: 3-octanol (OCT) and 4-methylcyclohexanol (MCH). Because we could potentially predict preference (a single value) using numerous glomerular-odor predictors and had a limited number of observations (dozens), we used dimensionality reduction to hold down the number of comparisons we made. We computed the principal components (PCs) of the glomerulus-odor responses (in either ORNs or PNs) across individuals (*Figure 1G–I*, *Figure 1—figure supplements 3 and 8*) and fit linear models to predict the behavior of individual flies from their values on the odor response PCs. No PCs of ORN neural activity could linearly predict OCT-MCH preference beyond the level of shuffled controls (n = 35 flies) (*Figure 1K and L*). The best ORN PC model only predicted odor-vs.-odor behavior with a nominal *R*² of 0.031. In contrast, PC 2 of PN activity was a statistically significant predictor of odor preference, accounting for 15% of preference variance in a training set of 47 flies (p=0.0063; *Figure 2—figure supplement 1C*) and 31% of preference variance on test data of flies (p=0.0069; *Figure 2—figure supplement 1D*). These p-values remain significant at α=0.05 following a Bonferroni correction for five comparisons. Combined train/test statistics (*R*² = 0.20; p=0.0001) are presented in *Figure 1K and M*. Thus, idiosyncratic PN calcium predicts odor vs. odor preference.

We conducted a follow-up analysis to contextualize the finding of calcium PCs predicting odor preference with an *R*² of ~0.2. This value is lower than 1.0 due to at least two factors: (1) any nonlinearity in the relationship between calcium responses and behavior, and (2) sampling error in, and temporal instability of, behavior and calcium responses over the duration of the experiment. A lower bound on the latter can be estimated from the repeatability of behavioral measures over time (*Figure 1—figure supplement 1B–E*). We performed a statistical analysis to roughly estimate model performance if there were no sampling error or drift in the measurement of behavior and calcium

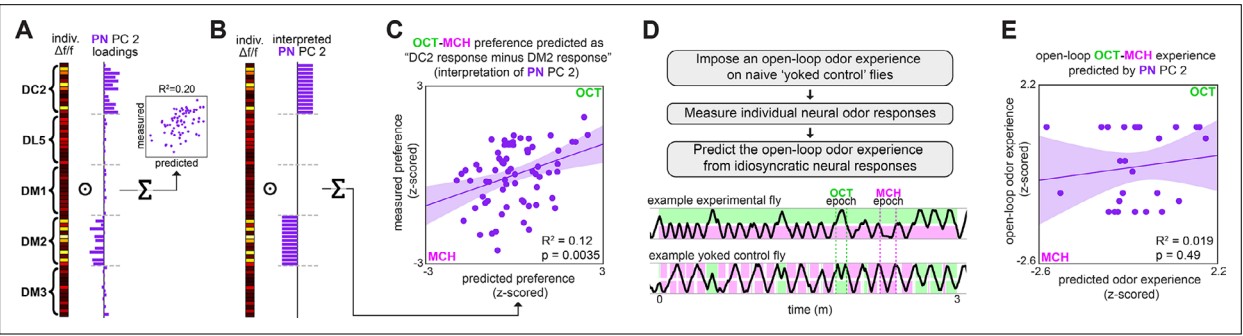

**Figure 2.** Variation in relative glomerular responses explains individual odor preference. (**A**) PC 2 loadings of projection neuron (PN) activity for flies tested for OCT-MCH preference (n = 69 flies). (**B**) Interpreted PN PC 2 loadings. (**C**) Measured OCT-MCH preference vs. preference predicted by the average peak PN response in DM2 minus DC2 across all odors (n = 69 flies). (**D**) Yoked-control experiment outline and example behavior traces. Experimental flies are free to move about tunnels permeated with steady-state OCT and MCH flowing into either end. Yoked-control flies are delivered the same odor at both ends of the tunnel that matches the odor experienced at the nose of the experimental fly at each moment in time. (**E**) Imposed odor experience vs. the odor experience predicted from PC 2 of PN activity (n = 27 flies) evaluated on the model trained from data in *Figure 1M*. Shaded regions in (**C, E**) are the 95% CIs of the fit estimated by bootstrapping.

The online version of this article includes the following figure supplement(s) for figure 2:

**Figure supplement 1.** Measured preference vs. projection neuron (PN) activity-based predicted preference, split by training/testing set.

**Figure supplement 2.** Time-dependent preference- and odor-decoding.

**Table 1.** Calcium and Brp-Short – behavior model statistics.

| Behavior measured | Neural predictor | Figure panel | n | $\beta_0$ | $\beta_1$ | $R^2$ | p-Value |
|---|---|---|---|---|---|---|---|
| OCT vs. AIR | PN calcium PC 1 | *Figure 2—figure supplement 1A* | 18 | –0.26 | –0.079 | 0.16 | 0.099 |
| OCT vs. AIR | PN calcium average all dimensions | *Figure 1—figure supplement 10I* | 53 | –0.051 | –0.38 | 0.098 | 0.022 |
| OCT vs. AIR | ORN calcium PC 1 | *Figure 1—figure supplement 10B* | 30 | –0.29 | –0.053 | 0.23 | 0.007 |
| OCT vs. AIR | ORN calcium average all dimensions | *Figure 1—figure supplement 10E* | 30 | –0.032 | –0.71 | 0.25 | 0.005 |
| OCT vs. MCH | PN calcium PC 2 | *Figure 2—figure supplement 1C* | 47 | –0.058 | –0.081 | 0.15 | 0.006 |
| OCT vs. MCH | PN calcium DM2–DC2 (% difference) | *Figure 2I* | 69 | –0.032 | –0.0018 | 0.12 | 0.004 |
| OCT vs. MCH | ORN calcium PC 1 | *Figure 1L* | 35 | –0.14 | –0.027 | 0.031 | 0.32 |
| OCT vs. MCH | ORN Brp-Short PC 2 (train data only) | *Figure 3—figure supplement 1I* | 22 | –0.087 | 0.017 | 0.22 | 0.028 |
| OCT vs. MCH | ORN Brp-Short PC 2 (all data) | *Figure 3F* | 53 | –0.019 | 0.012 | 0.088 | 0.031 |

MCH, 4-methylcyclohexanol; OCT, 3-octanol; ORN, olfactory receptor neuron; PC, principal component; PN, projection neuron.

responses (*Figure 1—figure supplement 9*; 'Materials and methods'). This analysis suggests that the measured correlation between calcium and behavior ($R^2_{latent}$) would be 0.46 in the absence of sampling error and temporal instability, but the uncertainty in this estimate is high (90% CI 0.06–0.90).

We additionally assessed the extent to which idiosyncratic calcium responses in ORNs or PNs could predict preference between air and a single aversive odor (OCT). We found a suggestive correlate: PC 1 of ORN calcium responses explained 23% of preference variance (n = 30 flies, p=0.0099, *Figure 1—figure supplement 10B*), but this association was dominated by a single outlier ($R^2$ of 0.078, p=0.14 with the outlier removed).

We next sought a biological understanding of the models associating calcium responses with odor preference. The loadings of the ORN and PN PCs indicate that variation across individuals was correlated at the level of glomeruli much more strongly than odorant (*Figure 1H*, *Figure 1—figure supplements 3 and 8*). This suggests that stochastic variation in the olfactory circuit results in individual-level fluctuations in the responses of glomeruli-specific rather than odor-specific responses. In the odor-vs.-odor preference model, the loadings of PC2 of PN calcium responses contrast the responses of the DM2 and DC2 glomeruli with opposing weights (*Figure 2A*), suggesting that the activation of DM2 relative to DC2 predicts the likelihood of a fly preferring OCT to MCH. Indeed, a linear model constructed from the average DM2 minus average DC2 PN response (*Figure 2B*) showed a statistically significant correlation with preference for OCT vs. MCH ($R^2 = 0.12$; p=0.0035; *Figure 2C*). The model slope coefficient was negative (*Table 1*), indicating that greater activation of DM2 vs. DC2 correlates with preference for MCH. With respect to odor-vs.-odor behavior, we conclude that the relative responses of DM2 vs. DC2 in PNs compactly predict an individual's preference.

Odor experience has been shown to modulate subsequent AL responses (*Golovin and Broadie, 2016*; *Iyengar et al., 2010*; *Sachse et al., 2007*). This raises the possibility that our models were actually predicting individual flies' past odor experiences (i.e., the specific pattern of odor stimulation flies received in the behavioral assay) rather than their preferences. To address this, we imposed the specific odor experiences of previously tracked flies (in the odor-vs.-odor assay) on naive 'yoked'

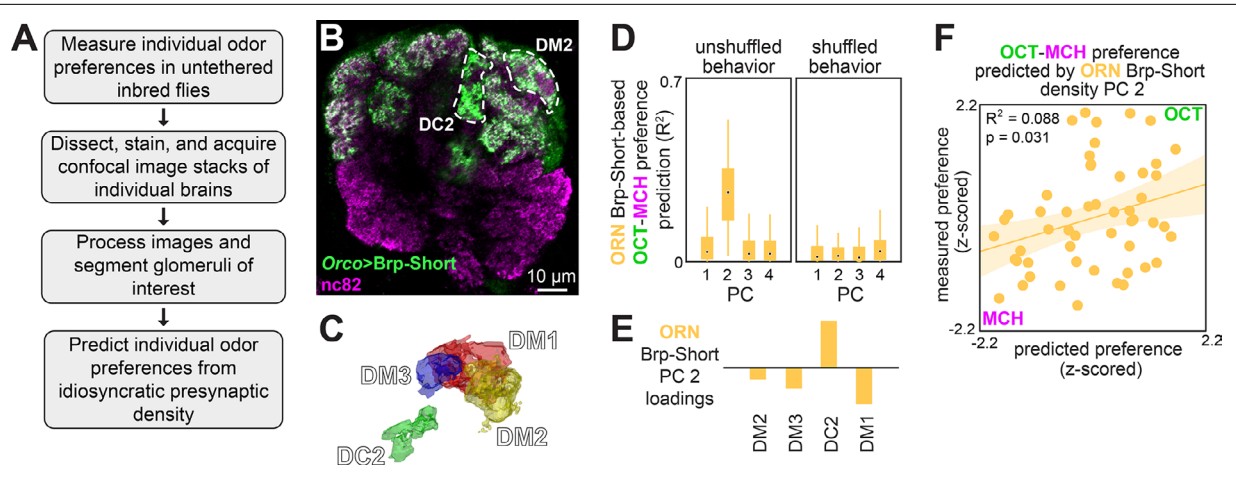

**Figure 3.** Idiosyncratic presynaptic marker density in DM2 and DC2 predicts OCT-MCH preference. (**A**) Experiment outline. (**B**) Example slice from a z-stack of the antennal lobe expressing Orco>Brp-Short (green) with DC2 and DM2 visible (white dashed outline). nc82 counterstain (magenta). (**C**) Example glomerulus segmentation masks extracted from an individual z-stack. (**D**) Bootstrapped $R^2$ of OCT-MCH preference prediction from each of the first four principal components of Brp-Short density measured in olfactory receptor neurons (ORNs) (training set, n = 22 flies). Points represent the median value, boxes represent the interquartile range, and whiskers the range of the data. (**E**) PC 2 loadings of Brp-Short density. (**F**) Measured OCT-MCH preference vs. preference predicted from PC 2 of ORN Brp-Short density in n = 53 flies using a model trained on a training set of n = 22 flies (see *Figure 3—figure supplement 1* for train/test flies analyzed separately).

The online version of this article includes the following figure supplement(s) for figure 3:

**Figure supplement 1.** ORN>Brp-Short characterization and model predictions.

**Figure supplement 2.** Calcium and Brp-Short predictor variation.

control flies (*Figure 2D*) and measured PN odor responses of the yoked flies. Applying the PN PC 2 model to the yoked calcium responses did not predict flies' odor experience ($R^2 = 0.019$, p=0.49; *Figure 2E*). This is consistent with PN calcium responses predicting odor preference rather than odor experience.

*Mazor and Laurent, 2005* found that PN response transients, rather than fixed points, contain more odor identity information. We therefore asked at which times during odor presentation an individual's neural responses could best predict odor preference. Applying our calcium-to-behavior models (PN PC2-odor-vs.-odor, as well as ORN PC1-odor-vs.-air, PN PC1-odor-vs.-air) to the time-varying calcium signals, we found that in all cases behavior prediction rose during odor delivery (*Figure 2—figure supplement 2*). In ORNs, the predictive accuracy remained high after odor offset, whereas in PNs it declined. The times during which calcium responses predicted individual behavior generally aligned to the times during which a linear classifier could decode odor identity from neuronal responses (*Figure 2—figure supplement 2D*), suggesting that idiosyncrasies in odor encoding predict individual preferences.

## Variation in a presynaptic scaffolding protein predicts odor preference

We next investigated how structural variation in the nervous system might relate to idiosyncratic behavior. Because PN, but not ORN, calcium responses predicted odor-vs.-odor preference, we hypothesized that a circuit element between ORNs to PNs could confer onto PNs behaviorally relevant physiological idiosyncrasies absent in ORNs. We therefore imaged presynaptic T-bar density in ORNs using transgenic mStrawberry-tagged Brp-Short, immunohistochemistry and confocal microscopy (*Mosca and Luo, 2014*) after measuring individual preference for OCT vs. MCH (*Figure 3A*). Brp-Short density was quantified as total fluorescence intensity/glomerulus volume for four of the five focus glomeruli (*Figure 3B*, *Figure 3—figure supplement 1A–F*; DL5 was not readily segmentable in our confocal samples). We chose this metric as we found it could be used to predict individual behavioral biases in a previous study (*Skutt-Kakaria et al., 2019*). This measure was consistent across hemispheres (*Figure 3—figure supplement 1C*), while also showing variation among individuals, like calcium responses.

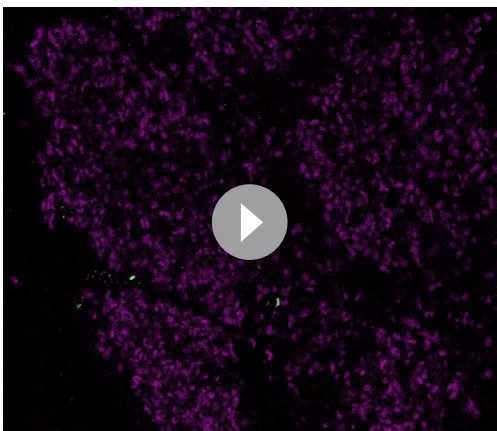

**Video 3.** Confocal image stack of expanded DC2>Brp-Short. Magenta is nc82 stain, green is Or13a>Brp-Short. Frames are z-slices spaced at 0.54 μm. Image height corresponds to a post-expansion field of view of 107 × 90 μm (~2.5× linear expansion factor).
https://elifesciences.org/articles/90511/figures#video3

To relate presynaptic structural variation and behavior, we used the same analytical approach as we had for calcium responses. PCs 1 and 2 of Brp-Short density had notable similarities to those of the calcium responses: PC 1 was positive across glomeruli and PC 2 exhibited a sign contrast between DC2 loadings and all other glomerulus loadings (*Figure 3—figure supplement 1G*). As in the PN calcium response models, PC 2 of Brp-Short density was the best predictor of odor-vs.-odor preferences in training data (*Figure 3D and E*, *Figure 3—figure supplement 1I*, $R^2$ = 0.22, n = 22 flies, p=0.028) and for test data (*Figure 3—figure supplement 1J*, $R^2$ = 0.078, n = 31 flies, p=0.13; statistics from combined train and test data: $R^2$ = 0.088, n = 53 flies, p=0.031; *Figure 3F*; median $R^2_{latent}$ 0.15, 90% CI 0.00–0.74). To better understand the microstructural basis of our Brp-Short density metric, we performed paired behavior and expansion microscopy (*Asano et al., 2018*; *Gao et al., 2019*) in flies expressing Brp-Short specifically in DC2-projecting ORNs (*Video 3*). Expansion yielded an approximately fourfold increase in linear resolution, allowing imaging of individual Brp-Short puncta (*Figure 3—figure supplement 1K*). While the sample size (n = 8) of this imaging pipeline was insufficient for a formal statistical analysis, the trend between Brp-Short density in DC2 (measured as individual puncta/glomerular volume) and odor-vs.-odor preference was more suggestive of a correlation than other metrics, such as median puncta volume (*Figure 3—figure supplement 1L and M*).

The best presynaptic density models are less predictive of behavior than the best calcium response models ($R^2$ = 0.088 vs. $R^2$ = 0.22; $R^2_{latent}$ 0.15 and 0.46, respectively; *Figure 2—figure supplement 1C and D* vs. *Figure 3—figure supplement 1I and J*), suggesting that presynaptic density variation is not the full explanation of calcium response variability. Nevertheless, differences in presynaptic inputs to PNs may contribute to variation in the calcium dynamics of those neurons, in turn giving rise to individual preferences for OCT vs. MCH.

## Developmental stochasticity in a simulated AL recapitulates empirical PN response variation

Finally, we sought an integrative understanding of how synaptic variation plays out across the olfactory circuit to produce behaviorally relevant physiological variation. We developed a leaky-integrate-and-fire model of the entire AL, comprising 3062 spiking neurons and synaptic connectivity taken directly from the *Drosophila* hemibrain connectome (*Scheffer et al., 2020*). After tuning the model to perform canonical AL computations, we introduced different kinds of stochastic variations to the circuit and determined which (if any) produce the patterns of idiosyncratic PN response variation observed in our calcium imaging experiments (*Figure 4A*). This approach assesses potential mechanisms linking developmental variation in synapses to physiological variation that apparently drives behavioral individuality.

The biophysical properties of neurons in our model (*Figure 4B*, *Table 2*) were determined by published electrophysiological studies (see 'Voltage model' in 'Materials and methods') and were similar to those used in previous fly models (*Kakaria and de Bivort, 2017*; *Pisokas et al., 2020*). The polarity of neurons was determined largely by their cell type (ORNs are excitatory, PNs predominantly excitatory, and LNs predominantly inhibitory – explained further in 'Materials and methods'). The strength of synaptic connections between any pair of AL neurons was given by the hemibrain connectome (*Scheffer et al., 2020*; *Figure 4C*). Odor inputs were simulated by injecting current into ORNs to produce spikes in those neurons at rates that match published ORN-odor recordings (*Münch and Galizia, 2016*), and the output of the system was recorded as the firing rates of PNs during

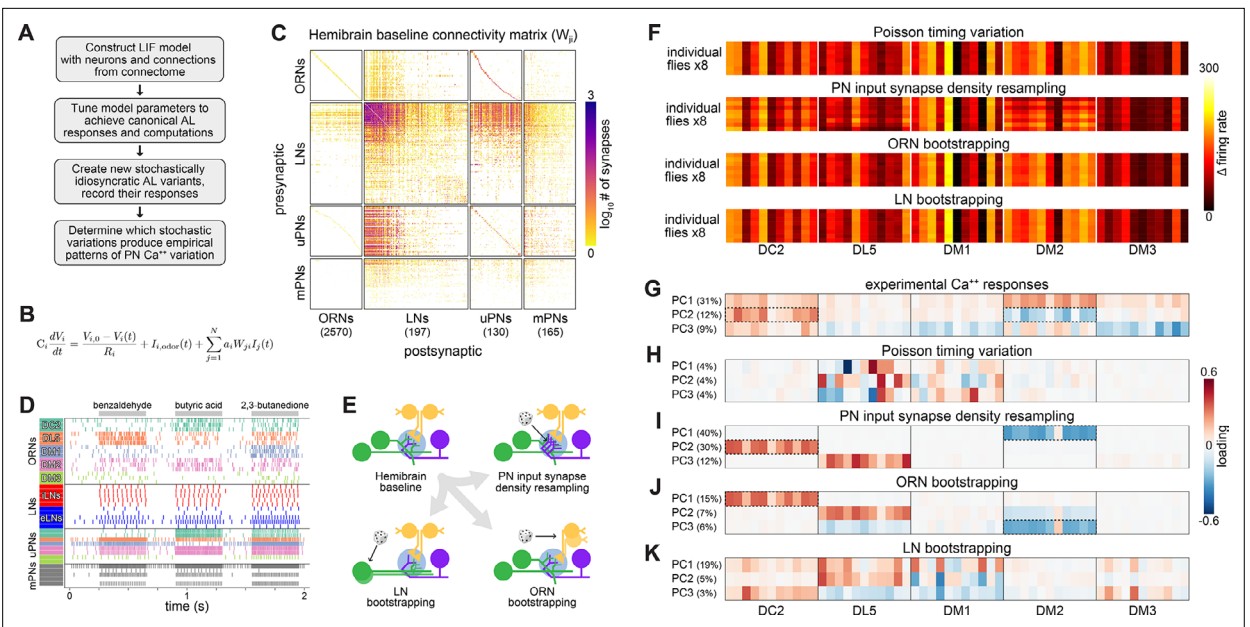

**Figure 4.** Simulation of olfactory circuits under developmental stochasticity. (**A**) Antennal lobe (AL) modeling analysis outline. (**B**) Leaky-integrator dynamics of each simulated neuron. When a neuron's voltage reaches its firing threshold, a templated action potential is inserted, and downstream neurons receive a postsynaptic current. See 'Antennal lobe modeling' in 'Materials and methods'. (**C**) Synaptic weight connectivity matrix, derived from the hemibrain connectome (*Scheffer et al., 2020*). (**D**) Spike raster for randomly selected example neurons from each AL cell type. Colors indicate olfactory receptor neuron (ORN)/projection neuron (PN) glomerular identity and LN polarity (i = inhibitory, e = excitatory). (**E**) Schematic illustrating sources of developmental stochasticity as implemented in the simulated AL framework. See *Video 4* for the effects of these resampling methods on the synaptic weight connectivity matrix. (**F**) PN glomerulus-odor response vectors for eight idiosyncratic ALs subject to Input spike Poisson timing variation, PN input synapse density resampling, and ORN and LN population bootstrapping. (**G**) Loadings of the principal components of PN glomerulus-odor responses as observed across experimental flies (top). Dotted outlines highlight loadings selective for the DC2 and DM2 glomerular responses, which underlie predictions of individual behavioral preference. (**H–K**) As in (**G**) for simulated PN glomerulus-odor responses subject to Input spike Poisson timing variation, PN input synapse density resampling, and ORN and LN population bootstrapping. See *Figure 4—figure supplement 5* for additional combinations of idiosyncrasy methods. In (**F–K**) the sequence of odors within each glomerular block is: OCT, 1-hexanol, ethyl-lactate, 2-heptanone, 1-pentanol, ethanol, geranyl acetate, hexyl acetate, MCH, pentyl acetate, and butanol.

The online version of this article includes the following figure supplement(s) for figure 4:

**Figure supplement 1.** Antennal lobe (AL) model raster plot.

**Figure supplement 2.** Antennal lobe (AL) model baseline outputs compared to experimental data.

**Figure supplement 3.** Sensitivity analysis of $a_{ORN}$, $a_{eLN}$, $a_{iLN}$, $a_{PN}$ parameters.

**Figure supplement 4.** Synapse counts vs. glomerular volume in the hemibrain and antennal lobe (AL) model.

**Figure supplement 5.** Projection neuron (PN) response PCA loadings under various sources of circuit idiosyncrasy.

**Figure supplement 6.** Classifiability of simulated idiosyncratic behavior under different sources of circuit idiosyncrasy.

**Table 2.** Typical electrophysiology features of antennal lobe cell types, used as model parameters.

| Parameter | Olfactory receptor neurons | Local neurons | Projection neurons |
| --- | --- | --- | --- |
| Membrane resting potential | –70 mV (*Dubin and Harris, 1997*) | –50 mV (*Seki et al., 2010*) | –55 mV (*Jeanne and Wilson, 2015*) |
| Action potential threshold | –50 mV (*Dubin and Harris, 1997*) | –40 mV (*Seki et al., 2010*) | –40 mV (*Jeanne and Wilson, 2015*) |
| Action potential minimum | –70 mV (*Cao et al., 2016*) | –60 mV (*Seki et al., 2010*) | –55 mV (*Jeanne and Wilson, 2015*) |
| Action potential maximum | 0 mV (*Dubin and Harris, 1997*) | 0 mV (*Seki et al., 2010*) | –30 mV (*Wilson and Laurent, 2005*) |
| Action potential duration | 2 ms (*Jeanne and Wilson, 2015*) | 4 ms (*Seki et al., 2010*) | 2 ms (*Jeanne and Wilson, 2015*) |
| Membrane capacitance | 73 pF (assumed = projection neurons) | 64 pF (*Huang et al., 2018*) | 73 pF (*Huang et al., 2018*) |
| Membrane resistance | 1.8 GOhm (*Dubin and Harris, 1997*) | 1 GOhm (*Seki et al., 2010*) | 0.3 GOhm (*Jeanne and Wilson, 2015*) |

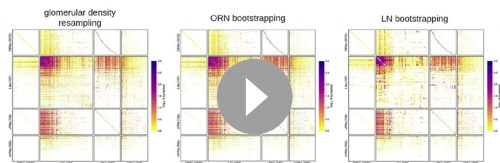

**Video 4.** Simulated antennal lobe (AL) connectivity matrices. Left: glomerular density resampling. Each frame corresponds to the hemibrain connectome synaptic weights, rescaled according to a sample from the relationship between synapse count and volume parameterized in *Figure 4—figure supplement 4*. Middle: olfactory receptor neuron (ORN) bootstrapping. Each frame corresponds to the hemibrain connectome synaptic weights, but with the population of ORNs projecting to each glomerulus resampled with replacement. Right: local neuron (LN) bootstrapping. Each frame corresponds to the hemibrain connectome synaptic weights, but with the population of LNs resampled with replacement.
https://elifesciences.org/articles/90511/figures#video4

odor stimulation (*Figure 4D*). At this point, there remained only four free parameters in our model, the relative sensitivity (postsynaptic current per upstream action potential) of each AL cell type (ORNs, PNs, excitatory LNs, and inhibitory LNs). We explored this parameter space manually and identified a configuration in which AL simulation (*Figure 4—figure supplement 1*) recapitulated four canonical properties seen experimentally (*Figure 4—figure supplement 2*): (1) typical firing rates at baseline and during odor stimulation (*Bhandawat et al., 2007*; *Dubin and Harris, 1997*; *Jeanne and Wilson, 2015*; *Seki et al., 2010*), (2) a more uniform distribution of PN firing rates compared to ORN rates (*Bhandawat et al., 2007*), (3) greater separation of PN odor representations compared to ORN representations (*Bhandawat et al., 2007*), and (4) a sublinear transfer function between ORNs and PNs (*Bhandawat et al., 2007*). Thus, our simulated AL appeared to perform the fundamental computations of real ALs, providing a baseline for assessing the effects of idiosyncratic variation.

We simulated stochastic individuality in the AL circuit in two ways (*Figure 4E*): (1) glomerular-level variation in PN input-synapse density (reflecting a statistical relationship observed between glomerular volume and synapse density in the hemibrain, *Figure 4—figure supplement 4*), and (2) bootstrapping of neuronal compositions within cell types (reflecting variety in developmental program outcomes for ORNs, PNs, etc.). *Video 4* shows the diverse connectivity matrices attained under these resampling approaches. We simulated odor responses in thousands of ALs made idiosyncratic by these sources of variation, and in each, recorded the firing rates of PNs when stimulated by the 12 odors from our experimental panel (*Figure 4F*, *Figure 4—figure supplement 1*).

To determine which sources of variation produced patterns of PN coding variation consistent with our empirical measurements, we compared principal components of PN responses from real idiosyncratic flies to those of simulated idiosyncratic ALs. Empirical PN responses are strongly correlated at the level of glomeruli (*Figure 4G*; *Figure 1—figure supplement 8*). As a positive control that the model can recapitulate this empirical structure, we resampled PN input-synapse density across glomeruli, producing PN response correlations strongly organized by glomerulus (*Figure 4I*). As a negative control, variation in PN responses due solely to Poisson timing of ORN input spikes (i.e., absent any circuit idiosyncrasy) was not organized at the glomerular level (*Figure 4H*). Strikingly, bootstrapping ORN population compositions yielded a strong glomerular organization in PN responses (*Figure 4J*). The loadings of the top PCs under ORN bootstrapping are dominated by responses of a single glomerulus to all odors, including DM2 and DC2. This is reminiscent of PC2 of PN calcium responses, with prominent (opposite sign) loadings for DM2 and DC2. Bootstrapping LNs, in contrast, produced much less glomerular organization (*Figure 4K*), with little resemblance to the loadings of the empirical calcium PCs. The PCA loadings for simulated PN responses under all combinations of cell type bootstrapping and PN input-synapse density resampling are given in *Figure 4—figure supplement 5*.

DM2 and DC2 (also DL5) stand out in the PCA loadings under PN input-synapse density resampling and ORN bootstrapping (*Figure 4I and J*), suggesting that behaviorally-relevant PN coding variation is recapitulated in this modeling framework. To formalize this analysis, for each idiosyncratic AL, we computed a 'behavioral preference' by applying the PN PC2 linear model (*Figure 1K and M*) to simulated PN responses. We then determined how accurately a linear classifier could distinguish OCT- vs. MCH-preferring ALs in the space of the first three PCs of PN responses (*Figure 4—figure supplement 6*). High accuracy was attained under PN input-synapse density resampling and ORN bootstrapping (sources of circuit variation that produced PN response loadings highlighting DM2 and DC2). Thus,

developmental variability in ORN populations may drive patterns of PN physiological variation that in turn drive individuality in odor-vs.-odor choice behavior.

## Discussion

We found an element of the *Drosophila* olfactory circuit in which individual patterns of physiological activity predict individual behavioral preferences. This circuit element can be considered a locus of individuality as it appears to contribute to idiosyncratic preferences among isogenic animals reared in the same environment. Specifically, the difference in the activation of PNs in DC2 and DM2 during odor exposure predicts idiosyncratic OCT-vs.-MCH preferences (*Figures 1 and 2*). This circuit element is in the olfactory sensory periphery and explains a large portion of the individuality signal, suggesting that behavioral idiosyncrasy arises early and suddenly in the sensorimotor transformation.

Correlating behavior to microscopic circuit features at the individual level is challenging (*Koulakov et al., 2005*). Measurements of both calcium responses and preference behavior are noisy. Calcium recordings are slow to acquire, making it hard to achieve sample sizes sufficient for machine-learning discovery of correlations with behavior. We conducted three major experiments (paired odor-vs.-odor preference and calcium recordings, odor-vs.-air preference and calcium recordings, and odor-vs.-odor and Brp-Short imaging), each with training and test sets on the scale of 20–60 individuals each. This allowed us to do some limited statistical discovery of correlations, which we restrained by conducting at most five exploratory correlation measurements between circuit and behavioral measures. We were particularly struck by the extent to which PN activity could predict preference between two aversive odors. Importantly, we confirmed this by evaluating the PN calcium–behavior model on a test set of flies measured several weeks after the training flies, finding the same statistically robust trend in both data partitions (training set: $R^2$ = 0.15, n = 47, p=0.0063; testing set: $R^2$ = 0.31, n = 22, p=0.0069; *Figure 2—figure supplement 1*).

Previous work has found mammalian peripheral circuit areas are predictive of individual behavior (*Britten et al., 1996*; *Morais et al., 2017*; *Newsome et al., 1989*; *Osborne et al., 2005*), but this study is among the first (*Linneweber et al., 2020*; *Mellert et al., 2016*; *Skutt-Kakaria et al., 2019*) to link cellular-level circuit variants and individual behavior in the absence of genetic variation. Another key conclusion is that loci of individuality are likely to vary, even within the sensory periphery, with the specific behavioral paradigm (i.e., odor-vs.-odor or odor-vs.-air). Our ability to predict behavioral preferences was limited by the repeatability of the behavior itself (*Figure 1—figure supplement 9*). Low persistence of odor preference may be attributable to factors like internal states or plasticity. It may be fruitful in future studies to map circuit elements whose activity predicts trial-to-trial behavioral fluctuations within individuals.

Seeking insight into the molecular basis of behaviorally relevant physiological variation, we imaged Brp in the axon terminals of the ORN-PN synapse using confocal and expansion microscopy. Brp glomerular density was a significant predictor of individual odor-vs.-odor preferences (*Figure 3*). The strongest predictor of OCT-MCH preference among principal components of Brp-Short density features contrastive loadings between DM2 and other glomeruli, similar to the DM2-DC2 contrast present in the model that predicts odor preference from PN calcium. This is consistent with the recent finding of a linear relationship between synaptic density and excitatory postsynaptic potentials (*Liu et al., 2022*) and another study in which idiosyncratic synaptic density in central complex output neurons predicts individual locomotor behavior (*Skutt-Kakaria et al., 2019*). The predictive relationship between Brp and behavior was weaker than that of PN calcium responses, suggesting there are other determinants, such as other synaptic proteins, neurite morphology, or the influence of idiosyncratic LNs (*Chou et al., 2010*) modulating the ORN-PN transformation (*Nagel et al., 2015*).

To integrate our synaptic and physiological results, we implemented a spiking model with 3062 neurons and synaptic weights drawn directly from the fly connectome (*Scheffer et al., 2020*; *Figure 4*). With light parameter tuning, this model recapitulated canonical AL computations, providing a baseline for assessing the effects of idiosyncratic stochastic variation. The apparent variation in odor responses across simulated individuals (*Figure 4F*) is less than that seen in the empirical calcium responses (*Figure 1H*), likely due to (1) biological phenomena missing from the model, (2) the lack of measurement noise, and (3) the fact that our perturbations are applied to the connectome of a single fly. When examining PCA loadings, however, simulating idiosyncratic ALs by varying PN input synapse density or bootstrapping ORNs produced correlated PN responses across odors in DC2 and

DM2, matching our experimental results. These sources of variation specifically implicate the ORN-PN synapse (like our Brp results) as an important substrate for establishing behaviorally relevant patterns of PN response variation.

The flies used in our experiments were isogenic and reared in standardized laboratory conditions that produce reduced behavioral individuality compared to enriched environments (*Akhund-Zade et al., 2019*; *Körholz et al., 2018*; *Zocher et al., 2020*). Yet, even these conditions yield substantial behavioral individuality. We do not expect variability in the expression of the flies' transgenes to be a major driver of this individuality as wildtype flies have a similarly broad distribution of odor preferences (*Honegger et al., 2020*). The ultimate source of stochasticity in this behavior remains a mystery, with possibilities ranging from thermal fluctuations at the molecular scale to macroscopic, but seemingly irrelevant, variations like the exact fill level of the culture media (*Honegger and de Bivort, 2018*). Developing nervous systems employ various compensation mechanisms to dampen out the effects of these fluctuations (*Marder, 2011*; *Tobin et al., 2017*). Behavioral variation may be beneficial, supporting a bet-hedging strategy (*Hopper, 1999*) to counter environmental fluctuations (*Akhund-Zade et al., 2020*; *Honegger et al., 2020*; *Kain et al., 2015*; *Krams et al., 2021*). Empirically, the net effect of dampening systems and accreted ontological fluctuations is individuals with diverse behaviors (*Gomez-Marin and Ghazanfar, 2019*). This process unfolds across all levels of biological regulation. Just as PN response variation appears to be partially rooted in glomerular Brp variation, the latter has its own causal roots, including, perhaps, stochasticity in gene expression (*Li et al., 2017*; *Raj et al., 2010*), itself a predictor of idiosyncratic behavioral biases (*Werkhoven et al., 2021*). Improved methods to longitudinally assay the fine-scale molecular and anatomical makeup of behaving organisms throughout development and adulthood will be invaluable to further illuminate the mechanistic origins of individuality.

## Materials and methods

**Key resources table**

| Reagent type (species) or resource | Designation | Source or reference | Identifiers | Additional information |
|---|---|---|---|---|
| Genetic reagent (*Drosophila melanogaster*) | P{20XUAS-IVS-GCaMP6m}attP40 | Bloomington *Drosophila* Stock Center | RRID:BDSC_42748 | |
| Genetic reagent (*D. melanogaster*) | w[*]; P{w[+mC]=Or13a-GAL4.F}40.1 | Bloomington *Drosophila* Stock Center | RRID:BDSC_9945 | |
| Genetic reagent (*D. melanogaster*) | w[*]; P{w[+mC]=Or19a-GAL4.F}61.1 | Bloomington *Drosophila* Stock Center | RRID:BDSC_9947 | |
| Genetic reagent (*D. melanogaster*) | w[*]; P{w[+mC]=Or22a-GAL4.7.717}14.2 | Bloomington *Drosophila* Stock Center | RRID:BDSC_9951 | |
| Genetic reagent (*D. melanogaster*) | w[*]; P{w[+mC]=Orco-GAL4.W}11.17; TM2/TM6B, Tb[1] | Bloomington *Drosophila* Stock Center | RRID:BDSC_26818 | |
| Genetic reagent (*D. melanogaster*) | isokh11 isogenic line | https://doi.org/10.1073/pnas.1901623116 | | *Honegger et al., 2020* |
| Genetic reagent (*D. melanogaster*) | GH146-Gal4 | https://doi.org/10.1073/pnas.1901623116 | | Gift of Y. Zhong (*Honegger et al., 2020*) |
| Genetic reagent (*D. melanogaster*) | w; UAS-Brp-Short-mStrawberry; UAS-mCD8-GFP; + | https://doi.org/10.7554/eLife.03726 | | Gift of T. Mosca (*Mosca and Luo, 2014*) |
| Antibody | Anti-nc82 (mouse monoclonal) | Developmental Studies Hybridoma Bank | DSHB:nc82; RRID:AB_2314866 | (1:40) |
| Antibody | Anti-GFP (chicken polyclonal) | Aves Labs | Aves Labs:GFP-1020; RRID:AB_10000240 | (1:1000) |
| Antibody | Anti-mStrawberry (rabbit polyclonal) | biorbyt | Biorbyt:orb256074 | (1:1000) |

*Continued on next page*

*Continued*

| Reagent type (species) or resource | Designation | Source or reference | Identifiers | Additional information |
|---|---|---|---|---|
| Antibody | Atto 647N-conjugated anti-mouse (goat polyclonal) | MilliporeSigma | Sigma-Aldrich:50185; RRID:AB_1137661 | (1:250) |
| Antibody | Alexa Fluor 568-conjugated anti-rabbit (goat polyclonal) | Thermo Fisher | Thermo Fisher Scientific:A-11011; RRID:AB_143157 | (1:250) |
| Antibody | Alexa Fluor 488-conjugated anti-chicken (goat polyclonal) | Thermo Fisher | Thermo Fisher Scientific:A-11039; RRID:AB_2534096 | (1:250) |
| Chemical compound, drug | 2-Heptanone | MilliporeSigma | CAS #110-43-0 | |
| Chemical compound, drug | 1-Pentanol | MilliporeSigma | CAS #71-41-0 | |
| Chemical compound, drug | 3-Octanol | MilliporeSigma | CAS #589-98-0 | |
| Chemical compound, drug | Hexyl-acetate | MilliporeSigma | CAS #142-92-7 | |
| Chemical compound, drug | 4-Methylcyclohexanol | MilliporeSigma | CAS #589-91-3 | |
| Chemical compound, drug | Pentyl acetate | MilliporeSigma | CAS #628-63-7 | |
| Chemical compound, drug | 1-Butanol | MilliporeSigma | CAS #71-36-3 | |
| Chemical compound, drug | Ethyl lactate | MilliporeSigma | CAS #97-64-3 | |
| Chemical compound, drug | Geranyl acetate | Millipore Sigma | CAS #105-87-3 | |
| Chemical compound, drug | 1-Hexanol | MilliporeSigma | CAS #111-27-34 | |
| Chemical compound, drug | Citronella java essential oil | Aura Cacia | Aura Cacia:191112 | |
| Software, algorithm | Python (version 3.6) | Python Software Foundation | RRID:SCR_008394 | |
| Software, algorithm | | MathWorks, *MATLAB pca documentation, 2018* | RRID:SCR_001622 | |

## Fly rearing

Experimental flies were reared in a *Drosophila* incubator (Percival Scientific DR-36VL) at 22°C, 40% relative humidity, and 12:12 hour light:dark cycle. Flies were fed cornmeal/dextrose medium, as previously described (*Honegger et al., 2020*). Mated female flies aged 3 days post-eclosion were used for behavioral persistence experiments. Mated female flies aged 7–15 days post-eclosion were used for all paired behavior-calcium imaging and immunohistochemistry experiments.

## Fly stocks

The following stocks were obtained from the Bloomington Drosophila Stock Center: P{20XUAS-IVS-GCaMP6m}attP40 (BDSC #42748), w[*]; P{w[+mC]=Or13a-GAL4.F}40.1 (BDSC #9945), w[*]; P{w[+mC]=Or19a-GAL4.F}61.1 (BDSC #9947), w[*]; P{w[+mC]=Or22a-GAL4.7.717}14.2 (BDSC #9951), w[*]; P{w[+mC]=Orco-GAL4.W}11.17; TM2/TM6B, Tb[1] (BDSC #26818). Transgenic lines were outcrossed to the isogenic line isokh11 (*Honegger et al., 2020*) for at least five generations prior to being used in any experiments. GH146-Gal4 was a gift provided by Y. Zhong (*Honegger et al., 2020*). w; UAS-Brp-Short-mStrawberry; UAS-mCD8-GFP;+ was a gift of Timothy Mosca and was not outcrossed to the isokh11 background (*Mosca and Luo, 2014*).

## Odor delivery

Odor delivery during behavioral tracking and neural activity imaging was controlled with isolation valve solenoids (NResearch Inc) (*Honegger et al., 2020*). Saturated headspace from 40 ml vials containing 5 ml pure odorant were serially diluted via carbon-filtered air to generate a variably (10–25%) saturated airstream controlled by digital flow controllers (Alicat Scientific) and presented to flies at total flow rates of ~100 ml/min. Dilution on the order of 10% is typical of other odor tunnel assays, as in *Claridge-Chang et al., 2009*. To yield the greatest signal of individual odor preference, dilution factors for odorants were adjusted on a week-by-week basis to ensure that the mean preference was approximately

50%. The odor panel used for imaging was comprised of the following odorants: 2-heptanone (CAS #110-43-0, MilliporeSigma), 1-pentanol (CAS #71-41-0, MilliporeSigma), 3-octanol (CAS #589-98-0, MilliporeSigma), hexyl-acetate (CAS #142-92-7, MilliporeSigma), 4-methylcyclohexanol (CAS #589-91-3, MilliporeSigma), pentyl acetate (CAS #628-63-7, MilliporeSigma), 1-butanol (CAS #71-36-3, MilliporeSigma), ethyl lactate (CAS #97-64-3, MilliporeSigma), geranyl acetate (CAS #105-87-3, MilliporeSigma), 1-hexanol (CAS #111-27-34, MilliporeSigma), citronella java essential oil (191112, Aura Cacia), and 200 proof ethanol (V1001, Decon Labs).

## Odor preference behavior

Odor preference was measured at 25°C and 20% relative humidity. As previously described (*Honegger et al., 2020*), individual flies confined to custom-fabricated tunnels were illuminated with infrared light and behavior was recorded with a digital camera (Basler) and zoom lens (Pentax). The odor choice tunnels were 50 mm long, 5 mm wide, and 1.3 mm tall. Custom real-time tracking software written in MATLAB was used to track centroid, velocity, and principal body axis angle throughout the behavioral experiment, as previously described (*Honegger et al., 2020*). After a 3-minute acclimation period, odorants were delivered to either end of the tunnel array for 3 minutes. Odor preference score was calculated as the fraction of time spent in the reference side of the tunnel during odor-on period minus the time spent in the reference side of the tunnel during the pre-odor acclimation period.

## Behavioral preference persistence measurements

After measuring odor preference, flies were stored in individual housing fly plates (modified 96-well plates; FlySorter, LLC) on standard food, temperature, humidity, and lighting conditions. Odor preference of the same individuals was measured 3 and/or 24 hours later. In some cases, fly tunnel position was randomized between measurements. Tunnel position had no apparent effect on preference persistence.

## Calcium imaging

Flies expressing GCaMP6m in defined neural subpopulations were imaged using a custom-built two-photon microscope and ultrafast Ti:Sapphire laser (Spectra-Physics Mai Tai) tuned to 930 nm, at a power of 20 mW out of the objective (Olympus XLUMPlanFL N ×20/1.00 W). For paired behavior and imaging experiments, the time elapsed between behavior measurement and imaging ranged from 15 minutes to 3 hours. Flies were anesthetized on ice and immobilized in an aluminum sheet with a female-fly-sized hole cut in it. The head cuticle between the antennae and ocelli was removed along with the tracheae to expose the ALs from the dorsal side. Volume scanning was performed using a piezoelectric objective mount (Physik Instrumente). ScanImage 2013 software (Vidrio Technologies) was used to coordinate galvanometer laser scanning and image acquisition. Custom MATLAB (Math-Works) scripts were used to coordinate image acquisition and control odor delivery. 256 × 192 (x–y) pixel 16-bit tiff images were recorded. The piezo travel distance was adjusted between 70 and 90 μm so as to cover most of the AL. The number of z-sections in a given odor panel delivery varied between 7 and 12 yielding a volume acquisition rate of 0.833 Hz. Odor delivery occurred from 6 to 9.6 s of each recording.

Each fly experienced up to four deliveries of the odor panel. The AL being recorded (left or right) was alternated after each successful completion of an odor panel. Odors were delivered in randomized order. In cases where baseline fluorescence was very weak or no obvious odor responses were visible, not all four panels were delivered.

## Glomerulus segmentation and labeling

Glomerular segmentation masks were extracted from raw image stacks using a *k*-means clustering algorithm based on time-varying voxel fluorescence intensities, as previously described (*Honegger et al., 2020*). Each image stack, corresponding to a single odor panel delivery, was processed individually. Time-varying voxel fluorescence values for each odor delivery were concatenated to yield a voxel-by-time matrix consisting of each voxel's recorded value during the course of all 13 odor deliveries of the odor panel. After z-scoring, principal component analysis was performed on this matrix and 75% of the variance was retained. Next, *k*-means (*k* = 80, 50 replicates with random starting seeds)

was performed to produce 50 distinct voxel cluster assignment maps that we next used to calculate a consensus map. This approach was more accurate than clustering based on a single *k*-means seed.

Of the 50 generated voxel cluster assignment maps, the top 5 were selected by choosing those maps with the lowest average within-cluster sum of distances, selecting for compact glomeruli. The remaining maps were discarded. Next, all isolated voxel islands in each of the top 5 maps were identified and pruned based on size (minimum size = 100 voxels, maximum size = 10,000 voxels). Finally, consensus clusters were calculated by finding voxel islands with significant overlap across all five of the pruned maps. Voxels that fell within a given cluster across all five pruned maps were added to the consensus cluster. This process was repeated for all clusters until the single consensus cluster map was complete. In some cases we found by manual inspection that some individual glomeruli were clearly split into two discrete clusters. These splits were remedied by automatically merging all consensus clusters whose centroids were separated by a physical distance of less than 30 voxels and whose peak odor response Spearman correlation was greater than 0.8. Finally, glomeruli were manually labeled based on anatomical position, morphology, and size (*Grabe et al., 2015*). We focused our analysis on five glomeruli (DM1, DM2, DM3, DL5, and DC2), which were the only glomeruli that could be observed in all paired behavior-calcium datasets. However, not all five glomeruli were identified in all recordings (*Figure 1—figure supplement 3*). Missing glomerular data was later mean-imputed. Using alternating least squares to impute data (running the pca function with option 'als' to infill missing values 1000 times and taking the mean infilled matrix – see Figure 1—figure supplement 5 of *Werkhoven et al., 2021*) had negligible effect on the fitted slope and predictive capacity of the PN PC2 OCT-MCH model compared to mean-infilling.

## Calcium image data analysis

All data was processed and analyzed in *MATLAB pca documentation, 2018* (MathWorks). Calcium responses for each voxel were calculated as $\Delta f/f = [f(t) - F]/F$, where $f(t)$ and $F$ are the instantaneous and average fluorescence, respectively. Each glomerulus' time-dependent calcium response was calculated as the mean $\Delta f/f$ across all voxels falling within the glomerulus' automatically-generated segmentation mask during a single volume acquisition. Time-varying odor responses were normalized to baseline by subtracting the median of pre-odor $\Delta f/f$ from each trace. Peak odor response was calculated as the maximum fluorescence signal from 7.2s to 10.8s (images 6–9) of the recording.

To compute principal components of calcium dynamics, each fly's complement of odor panel responses (a 5 glomeruli by 13 odors = 65-dimensional vector) was concatenated. Missing glomerulus-odor response values were filled in with the mean glomerulus-odor pair across all fly recordings for which the data was not missing. After infilling, principal component analysis was carried out with individual odor panel deliveries as observations and glomerulus-odor responses pairs as features.

Inter- and intra-fly distances (*Figure 1J*) were calculated using the projections of each fly's glomerulus-odor responses onto all principal components. For each fly, the average Euclidean distance between response projections (1) among left lobe trials, (2) among right lobe trials, and (3) between left and right lobe trials were averaged together to get a single within-fly distance. Intra-fly distances were computed in a similar fashion (for each fly, taking the average distance of its response projections to those of other flies using only left lobe trials/only right lobe trials/between left-right trials, then averaging these three values to get a single across-fly distance).

In a subset of experiments in which we imaged calcium activity, some solenoids failed to open, resulting in the failure of odor delivery in a small number of trials. In these cases, we identified trials with valve failures by manually recognizing that glomeruli failed to respond during the nominal odor period. These trials were treated as missing data and infilled, as described above. Fewer than ~10% of flies and 5% of odor trials were affected.

For all predictive models constructed, the average principal component score or glomerulus-odor $\Delta f/f$ response across trials was used per individual; that is, each fly contributed one data point to the relevant model. Linear models were constructed from behavior scores and the relevant predictor (principal component, average $\Delta f/f$ across dimensions, specific glomerulus measurements) as described in the text and *Tables 1 and 2*. All reported linear model p-values are nominal, that is, unadjusted for multiple hypothesis comparisons. 95% CIs around model regression lines were estimated as± SDs of the value of the regression line at each x-position across 2000 bootstrap replicates (resampling flies). To predict behavior as a function of time during odor delivery, we analyzed data as described above,

but considered only Δf/f at each single time point (*Figure 2—figure supplement 2A–C*), rather than averaging during the peak response interval.

To decode individual identity from neural responses, we first performed PCA on individual odor panel peak responses. We retained principal component scores constituting specified fractions of variance (*Figure 1—figure supplement 6A*) and trained a linear logistic classifier to predict individual identity from single-odor panel deliveries.

To decode odor identity from neural responses, each of the five recorded glomeruli were used as features, and the calcium response of each glomerulus to a specific odor at a specified time point were used as observations (PNs, n = 5317 odor deliveries; ORNs, n = 2704 odor deliveries). A linear logistic classifier was trained to predict the known odor identity using twofold cross-validation. That is, a model was trained on half the data and evaluated on the remaining half, and then this process was repeated with the train and test half reversed. The decoding accuracy was quantified as the fraction of odor deliveries in which the predicted odor was correct.

## Inference of correlation between latent calcium and behavior states

We performed a simulation-based analysis to infer the strength of the correlation between latent calcium (Brp) and behavior states, given the $R^2$ of a given linear model. *Figure 1—figure supplement 9* is a schematic of a possible data generation process that underlies our observed data. We assume that the 'true' behavioral and calcium values of the animal are captured by unobserved latent states $X_c$ and $X_b$, respectively, such that the $R^2$ between $X_c$ and $X_b$ is the biological signal captured by the model, having adjusted for the noise associated with actually measuring behavior and calcium ($R^2_{latent}$). Our calcium and odor preference scores are subject to measurement error and temporal instability (behavior and neural activity were measured 1–3 hours apart). These effects are both noise with respect to estimating the linear relationship between calcium and behavior. Their magnitude can be estimated using the empirical repeatability of behavior and calcium experiments respectively. Thus, our overall approach was to assume true latent behavior and calcium signals that are correlated by the level set at $R^2_{latent}$, add noise commensurate with the repeatability of these measures to simulate measured behavior and calcium, and record the simulated empirical $R^2$ between these measured signals. This was done many times to estimate distributions of empirical $R^2$ given $R^2_{latent}$. These distributions could finally be used in the inverse direction to infer $R^2_{latent}$ given the actual model $R^2$ values computed in our study.

Specifically, we simulated $X_c$ as a set of $N$ standard normal variables ($N$ equaling the number of flies used to compute a correlation between predicted and measured preference) and generated $X_b = r_{latent} X_c + [1 − (r_{latent})^2 Z]^{1/2}$, where $Z$ is a set of $N$ standard normal variables uncorrelated with $X_c$, a procedure that ensures that $corr(X_c, X_b) = r_{latent}$. Next, we simulated observed calcium readouts $X_c'$ and $X_c''$ such that $corr(X_c, X_c') = corr(X_c, X_c'') = r_c$. Similarly, we simulated noisy observed behavioral assay readouts $X_b'$ and $X_b''$, such that $corr(X_b, X_b') = corr(X_b, X_b'') = r_b$. The values of $r_c$ and $r_b$ were drawn from the empirical repeatability of calcium ($R^2_{c,c}$) and behavior ($R^2_{b,b}$) respectively as follows. Since calcium is a multidimensional measure, and our calcium model predictors are based on principal components of glomerulus-odor responses, we used variance explained along the PCs to calculate a single value for the calcium repeatability $R^2_{c,c}$. We compared the eigenvalues of the real calcium PCA to those of shuffled calcium data (shuffling glomerulus/odor responses for each individual fly), computing $R^2_{c,c}$ by summing the variance explained along the PCs of the calcium data up until the component-wise variance for the calcium data fell below that of the shuffled data, a similar approach as done in *Berman et al., 2014* and *Werkhoven et al., 2021*. That is, we determined which empirical PCs had more variance than their corresponding rank-matched PC in shuffled data, interpreted the remaining PCs as harboring the noise of the experiment, and totaled the variance explained of the non-noise PCs as our measure of the repeatability of the measurement as a whole. $R^2_{c,c}$ was calculated to be 0.77 for the full PN calcium data.

To incorporate uncertainty in calcium-calcium repeatability, we utilized bootstrapping. We resampled the calcium data associated with individual flies 10,000 times, performed PCA and computed $R^2_{c,c}$ for each resampled dataset, then set $r_c = (R^2_{c,c})^{1/4}$ to ensure $corr(X_c', X_c'')^2 = R^2_{c,c}$. For behavior–behavior uncertainty, we set $r_b$ from the repeatability across odor preference trials in the same flies measured 3 hours apart ($R^2_{b,b} = 0.12$ for OCT vs. MCH, *Figure 1—figure supplement 1D* using the full dataset of flies). We also resampled the flies 10,000 times, computed $R^2_{b,b}$ for each resampled dataset, and set $r_b = (R^2_{b,b})^{1/4}$ to ensure $corr(X_b', X_b'')^2 = R^2_{b,b}$.

We varied $r_{latent}$ from 0 to 1 in increments of 0.01, and for each $r_{latent}$ and bootstrap iteration we simulated a set of $N$ $X_c$, and generated $X_b$, $X_c'$, $X_c''$, $X_b'$, and $X_b''$, then we computed a simulated observed calcium–behavior relationship strength $R^2_{c,b} = corr(X_c', X_b')^2$. We repeated this simulation 10,000 times for each $r_{latent}$, transformed $r_{latent}$ to $R^2_{latent}$ such that for a quantile of interest $q$, $P(r_{latent} \leq q)$ matched $P(R^2_{latent} \leq q^2)$, and plotted the resultant relationship between $R^2_{latent}$ against $R^2_{c,b}$ (percentiles of $R^2_{c,b}$ are displayed in *Figure 1—figure supplement 9B*). We inferred $R^2_{latent}$ by first drawing bootstrapped samples of calcium–behavior $R^2$, then adding together the marginal distributions of $R^2_{latent}$ for each calcium–behavior $R^2$. We report the median $R^2_{latent}$ and 90% CI as estimated by the 5th–95th quantiles.

The procedure outlined above was done analogously for models using Brp-Short relative fluorescence intensity, performing the PCA-based calcium response repeatability step with PCA on the multidimensional Brp-Short relative fluorescence intensity (which yielded $R^2_{brp,brp} = 0.78$).

## DoOR data
DoOR data for the glomeruli and odors relevant to our study was downloaded from http://neuro.uni-konstanz.de/DoOR/default.html (*Münch and Galizia, 2016*).

## Yoked odor experience experiments
We selected six flies for which both odor preference and neural activity were recorded to serve as the basis for imposed odor experiences for yoked control flies. The experimental flies were chosen to represent a diversity of preference scores. Each experimental fly's odor experience was binned into discrete odor bouts to represent experience of either MCH or OCT based on its location in the tunnel as a function of time (*Figure 2D*). Odor bouts lasting less than 100 ms were omitted due to limitations on odor-switching capabilities of the odor delivery apparatus. To deliver a given experimental fly's odor experience to yoked controls, we set both odor streams (on either end of the tunnel apparatus) to deliver the same odor experienced by the experimental fly at that moment during the odor-on period. No odor was delivered to yoked controls during time points in which the experimental fly resided in the tunnel choice zone (central 5 mm). See *Figure 2D* for an example pair of experimental fly and yoked control behavior and odor experience.

## Immunohistochemistry
After measuring odor preference behavior, 7–15-day-old flies were anesthetized on ice and brains were dissected in phosphate-buffered saline (PBS). Dissection and immunohistochemistry were carried out as previously reported (*Wu and Luo, 2006*). The experimenter was blind to the behavioral scores of all individuals throughout dissection, imaging, and analysis. Individual identities were maintained by fixing, washing, and staining each brain in an individual 0.2 ml PCR tube using fluid volumes of 100 ul per brain (Fisher Scientific). Primary incubation solution contained mouse anti-nc82 (1:40, DSHB), chicken anti-GFP (1:1000, Aves Labs), rabbit anti-mStrawberry (1:1000, biorbyt), and 5% normal goat serum (NGS, Invitrogen) in PBT (0.5% Triton X-100 in PBS). Secondary incubation solution contained Atto 647N-conjugated goat anti-mouse (1:250, MilliporeSigma), Alexa Fluor 568-conjugated goat anti-rabbit (1:250), Alexa Fluor 488-conjugated goat anti-chicken (1:250, Thermo Fisher), and 5% NGS in PBT. Primary and secondary incubation times were two and three overnights, respectively, at 4°C. Stained samples were mounted and cleared in Vectashield (H-1000, Vector Laboratories) between two coverslips (12-568B, Fisher Scientific). Two reinforcement labels (5720, Avery) were stacked to create a 0.15 mm spacer.

## Expansion microscopy
Immunohistochemistry for expansion microscopy was carried out as described above, with the exception that antibody concentrations were modified as follows: mouse anti-nc82 (1:40), chicken anti-GFP (1:200), rabbit anti-mStrawberry (1:200), Atto 647N-conjugated goat anti-mouse (1:100), Alexa Fluor 568-conjugated goat anti-rabbit (1:100), and Alexa Fluor 488-conjugated goat anti-chicken (1:100). Expansion of stained samples was performed as previously described (*Asano et al., 2018*; *Gao et al., 2019*). Expanded samples were mounted in coverslip-bottom Petri dishes (MatTek Corporation) and anchored by treating the coverslip with poly-L-lysine solution (MilliporeSigma) as previously described (*Asano et al., 2018*).

## Confocal imaging

All confocal imaging was carried out at the Harvard Center for Biological Imaging. Unexpanded samples were imaged on an LSM700 (Zeiss) inverted confocal microscope equipped with a ×40 oil-immersion objective (1.3 NA, EC Plan Neofluar, Zeiss). Expanded samples were imaged on an LSM880 (Zeiss) inverted confocal microscope equipped with a ×40 water-immersion objective (1.1 NA, LD C-Apochromat, Zeiss). Acquisition of z-stacks was automated with Zen Black software (Zeiss).

## Standard confocal image analysis

We used custom semi-automated code to generate glomerular segmentation masks from confocal z-stacks of unexpanded Orco>Brp-Short brains. Using MATLAB, each image channel was median filtered ($\sigma_x$, $\sigma_y$, $\sigma_z$ = 11, 11, 1 pixels) and downsampled in $x$ and $y$ by a factor of 11. Next, an ORN mask was generated by multiplying and thresholding the Orco>mCD8 and Orco>Brp-Short channels. Next, a locally normalized nc82 and Orco>mCD8 image stack were multiplied and thresholded, and the ORN mask was applied to remove background and other undesired brain structures. This pipeline resulted in a binary image stack that maximized the contrast of the glomerular structure of the AL. We then applied a binary distance transform and watershed transform to generate discrete subregions that aimed to represent segmentation masks for each glomerulus tagged by Orco-Gal4.

However, this procedure generally resulted in some degree of under-segmentation; that is, some glomerular segmentation masks were merged. To split each merged segmentation mask, we convolved a ball (whose radius was proportional to the cube root of the volume of the segmentation mask in question) across the mask and thresholded the resulting image. The rationale of this procedure was that two merged glomeruli would exhibit a mask shape resembling two touching spheres, and convolving a similarly sized sphere across this volume followed by thresholding would split the merged object. After ball convolution, we repeated the distance and watershed transform to once more generate discrete subregions representing glomerular segmentation masks. This second watershed step generally resulted in over-segmentation; that is, by visual inspection it was apparent that many glomeruli were split into multiple subregions. Therefore, we finally manually agglomerated the over-segmented subregions to generate single segmentation masks for each glomerulus of interest. We used a published atlas to aid manual identification of glomeruli (*Grabe et al., 2015*). The total Brp-Short fluorescence signal within each glomerulus was determined and divided by the volume of the glomerulus' segmentation mask to calculate Brp-Short density values.

## Expansion microscopy image analysis

The spots function in Imaris 9.0 (Bitplane) was used to identify individual Brp-Short puncta in expanded sample image stacks of Or13a>Brp-Short samples (*Mosca and Luo, 2014*). The spot size was set to 0.5 um, background subtraction and region-growing were enabled, and the default spot quality threshold was used for each image stack. Identified spots were used to mask the Brp-Short channel and the resultant image was saved as a new stack. In MATLAB, a glomerular mask was generated by smoothing ($\sigma_x$, $\sigma_y$, $\sigma_z$ = 40, 40, 8 pixels) and thresholding (92.5th percentile) the raw Brp-Short image stack. The mask was then applied to the spot image stack to remove background spots. Finally, the masked spot image stack was binarized and spot number and properties were quantified.

## Antennal lobe modeling

We constructed a model of the AL to test the effect of circuit variation on PN activity variation across individuals. Our general approach to producing realistic circuit activity with the AL model was (1) using experimentally measured parameters whenever possible (principally the connectome wiring diagram and biophysical parameters measured electrophysiologically), (2) associating free parameters only with biologically plausible categories of elements, while minimizing their number, and (3) tuning the model using those free parameters so that it reproduced high-level patterns of activity considered in the field to represent the canonical operations of the AL. Simulations were run in Python (version 3.6) (*Rossum and Drake, 2011*), and model outputs were analyzed using Jupyter notebooks (*Kluyver et al., 2016*) and Python and MATLAB scripts.

## AL model neurons

Release 1.2 of the hemibrain connectomics dataset (*Scheffer et al., 2020*) was used to set the connections in the model. Hemibrain body IDs for ORNs, LNs, and PNs were obtained via the lists of neurons supplied in the supplementary tables in *Schlegel et al., 2020*. ORNs and PNs of non-olfactory glomeruli (VP1d, VP1l, VP1m, VP2, VP3, VP4, VP5) were ignored, leaving 51 glomeruli. Synaptic connections between the remaining 2574 ORNs, 197 LNs, 166 mPNs, and 130 uPNs were queried using the neuprint-python API (*Plaza et al., 2022*). All ORNs were assigned to be excitatory (*Wilson, 2013*). Polarities were assigned to PNs based on the neurotransmitter assignments in *Bates et al., 2020*. mPNs without neurotransmitter information were randomly assigned an excitatory polarity with probability equal to the fraction of neurotransmitter-identified mPNs that are cholinergic; the same process was performed for uPNs. After confirming that the model's output was qualitatively robust to which mPNs and uPNs were randomly chosen, this random assignment was performed once and then frozen for subsequent analyses.

Of the 197 LNs, we assigned 31 to be excitatory, based on the estimated 1:5.4 ratio of eLNs to iLNs in the AL (*Tsai et al., 2018*). To account for observations that eLNs broadly innervate the AL (*Shang et al., 2007*), all LNs were ranked by the number of innervated glomeruli, and the 31 eLNs were chosen uniformly at random from the top 50% of LNs in the list. This produced a distribution of glomerular innervations in eLNs qualitatively similar to that of *krasavietz* LNs in Supplementary Figure 6 of *Chou et al., 2010*.

## Voltage model

We used a single-compartment leaky-integrate-and-fire voltage model for all neurons as in *Kakaria and de Bivort, 2017*, in which each neuron had a voltage $V_i(t)$ and current $I_i(t)$. When the voltage of neuron *i* was beneath its threshold $V_{i,\,thr}$, the following dynamics were obeyed:

$$C_i \frac{dV_i}{dt} = \frac{V_{i,0} - V_i(t)}{R_i} + I_{i,odor}(t) + \sum_{j=1}^{N} a_i W_{ji} I_j(t)$$

Each neuron *i* had electrical properties: membrane capacitance $C_i$, resistance $R_i$, and resting membrane potential $V_{i,0}$ with values from electrophysiology measurements (*Table 2*).

When the voltage of a neuron exceeded the threshold $V_{i,\,thr}$, a templated action potential was filled into its voltage time trace, and a templated postsynaptic current was added to all downstream neurons, following the definitions in *Kakaria and de Bivort, 2017*.

Odor stimuli were simulated by triggering ORNs to spike at frequencies matching known olfactory receptor responses to the desired odor. The timing of odor-evoked spikes was given by a Poisson process, with firing rate *FR* for ORNs of a given glomerulus governed by

$$FR_{glom,odor}(t) = FR_{max} D_{glom,odor} \left( f_a + (1 - f_a) e^{-t/t_a} \right)$$

$FR_{max}$, the maximum ORN firing rate, was set to 400 Hz. $D_{glom,\,odor}$ is a value between 0 and 1 from the DoOR database, representing the response of an odorant receptor/glomerulus to an odor, estimated from electrophysiology and/or fluorescence data (*Münch and Galizia, 2016*). ORNs display adaptation to odor stimuli (*Wilson, 2013*), captured by the final term with timescale $t_a$ = 110 ms to 75% of the initial value, as done in *Kao and Lo, 2020*. Thus, the functional maximum firing rate of an ORN was 75% of 400 Hz = 300 Hz, matching the highest ORN firing rates observed experimentally (*Hallem et al., 2004*). After determining the times of ORN spikes according to this firing-rate rule, spikes were induced by the addition of $10^6$ picoamps in a single time step. This reliably triggered an action potential in the ORN, regardless of currents from other neurons. In the absence of odors, spike times for ORNs were drawn by a Poisson process at 10 Hz, to match reported spontaneous firing rates (*de Bruyne et al., 2001*).

For odor-glomeruli combinations with missing DoOR values (40% of the dataset), we performed imputation via alternating least squares using the pca function with option 'als' to infill missing values (MATLAB documentation) on the odor × glomerulus matrix 1000 times and taking the mean infilled matrix, which provides a closer match to ground truth missing values than a single run of ALS (*Figure 1—figure supplement 5* of *Werkhoven et al., 2021*).

A neuron $j$ presynaptic to $i$ supplies its current $I_j(t)$ scaled by the synapse strength $W_{ji}$, the number of synapses in the hemibrain dataset from neuron $j$ to $i$. Rows in $W$ corresponding to neurons with inhibitory polarity (i.e., GABAergic PNs or LNs) were set negative. Finally, postsynaptic neurons (columns of the connectivity matrix) have a class-specific multiplier $a_i$, a hand-tuned value, described below.

## AL model tuning

Class-specific multiplier current multipliers ($a_i$) were tuned using the panel of 18 odors from *Bhandawat et al., 2007* (our source for several experimental observations of high-level AL function): benzaldehyde, butyric acid, 2,3-butanedione, 1-butanol, cyclohexanone, Z3-hexenol, ethyl butyrate, ethyl acetate, geranyl acetate, isopentyl acetate, isoamyl acetate, 4-methylphenol, methyl salicylate, 3-methylthio-1-propanol, octanal, 2-octanone, pentyl acetate, E2-hexenal, trans-2-hexenal, and gamma-valerolactone. Odors were 'administered' for 400 ms each, with 300 ms odor-free pauses between odor stimuli.

The high-level functions of the AL that represent a baseline, working condition were (1) firing rates for ORNs, LNs, and PNs matching the literature (listed in *Table 2* and see *Bhandawat et al., 2007*; *Dubin and Harris, 1997*; *Jeanne and Wilson, 2015*; *Seki et al., 2010*), (2) a more uniform distribution of PN firing rates during odor stimuli compared to ORN firing rates, (3) greater separation of representations of odors in PN-coding space than in ORN-coding space, and (4) a sublinear transfer function between ORN firing rates and PN firing rates. Features (2)–(4) relate to the role of the AL in enhancing the separability of similar odors (*Bhandawat et al., 2007*).

To find a parameterization with those functions, we tuned the values of $a_i$ as scalar multipliers on ORN, eLN, iLN, and PN columns of the hemibrain connectivity matrix. Thus, these values represent cell type-specific sensitivities to presynaptic currents, which may be justified by the fact that ORNs/LNs/PNs are genetically distinct cell populations (*McLaughlin et al., 2021*; *Xie et al., 2021*). A grid search of the four class-wise sensitivity parameters produced a configuration that reasonably satisfied the above criteria (*Figure 4—figure supplement 2*). In this configuration, the ORN columns of the hemibrain connectivity matrix are scaled by 0.1, eLNs by 0.04, iLNs by 0.02, and PNs by 0.4. The relatively large multiplier on PNs is potentially consistent with the fact that PNs are sensitive to small differences between weak ORN inputs (*Bhandawat et al., 2007*). Model outputs were robust over several different sets of $a_i$, provided iLN sensitivity $\simeq$ eLN<ORN<PN.

We analyzed the sensitivity of the model's parameters around their baseline values of $a_{ORN}$, $a_{eLN}$, $a_{iLN}$, $a_{PN}$ = (0.1, 0.04, 0.02, 0.4). Each parameter was independently scaled up to 4× or 1/4× of its baseline value (*Figure 4—figure supplement 3*), and the PN firing rates recorded. Separately, multiple-parameter manipulations were performed by multiplying each parameter by a random log-Normal value with mean 1 and ±1 SD corresponding to a 2× or 0.5× scaling on each parameter. Mean PN-odor responses were calculated for all manipulated runs and compared to the mean PN-odor responses for the baseline configuration. A manipulation effect size was calculated by Cohen's $d$ ((mean manipulated response – mean baseline response)/(pooled standard deviation)). None of these manipulations reached effect size magnitudes larger than 0.9 (which can be roughly interpreted as the number of SDs in the baseline PN responses away from the mean baseline PN response), which signaled that the model was robust to the sensitivity parameters in this range. The most sensitive parameter was, unsurprisingly, $a_{PN}$.

Notable ways in which the model behavior deviates from experimental recordings (and thus caveats on the interpretation of the model) include (1) model LNs appear to have more heterogeneous firing rates than real LNs, with many LNs inactive for this panel of odor stimuli. This likely reflects a lack of plastic/homeostatic mechanisms in the model to regularize LN firing rates given their variable synaptic connectivity (*Chou et al., 2010*). (2) Some PNs had off-odor rates that are high compared to real PNs, resulting in a distribution of ON-OFF responses that had a lower limit than in real recordings. Qualitatively close matches were achieved between the model and experimental data in the distributions of odor representations in ORN vs. PN spaces and the nonlinearity of the ORN-PN transfer function.

## AL model circuit variation generation

We generated AL circuit variability in two ways: cell-type bootstrapping and synapse density resampling. These methods assume that the distribution of circuit configurations across individual ALs can

be generated by resampling circuit components within a single individual's AL (neurons and glomerular synaptic densities, respectively, from the hemibrain EM volume).

To test the effect of developmental variation in the complement of neurons of particular types, we bootstrapped populations of interest from the list of hemibrain neurons. Resampling with replacement of ORNs was performed glomerulus-by-glomerulus, that is, separately among each pool of ORNs expressing a particular *Odorant receptor* gene. The same was done for PNs. For LNs, all 197 LNs were treated as a single pool; there was no finer operation based on LN subtypes or glomerular innervations. This choice reflects the high developmental variability of LNs (*Chou et al., 2010*). The number of synapses between a pair of bootstrapped neurons was equal to the synapse count between those neurons in the hemibrain connectivity matrix.

In some glomeruli, bootstrapping PNs produced unreasonably high variance in the total PN synapse count. For instance, DP1m, DC4, and DM3 each harbor PNs that differ in total synapse count by a factor of ~10. Since these glomeruli have between two to three PNs each, in a sizable proportion of bootstrap samples, all-highly connected (or all-lowly) connected PNs are chosen in such glomeruli. To remedy this biologically unrealistic outcome, we examined the relationship between total input PN synapses within a glomerulus and glomerular volume (*Figure 4—figure supplement 4*). In the 'synapse density resampling' method, we required that the number of PN input synapses within a glomerulus reflect a draw from the empirical relationship between total input PN synapses and glomerular volume as present in the hemibrain dataset. This was achieved by, for each glomerulus, sampling from the following distribution that depends on glomerular volume, then multiplying the number of PN input synapses by a scalar to match that sampled value:

$$\log S_g = \log \left( a V_g^d \right) + \varepsilon_g, \varepsilon_g \sim N \left( 0, \sigma^2 \right)$$

Here, $S_g$ is the PN input synapse count for glomerulus $g$, $V_g$ is the volume of glomerulus $g$ (in cubic microns), $\varepsilon$ is a Gaussian noise variable with SD $\sigma$, and $a$, $d$ are the scaling factor and exponent of the volume term, respectively. The values of these parameters ($a = 8.98$, $d = 0.73$, $\sigma = 0.38$) were fit using maximum likelihood.

## Quantification and statistical analysis

All fly behavior and calcium data was processed and analyzed in *MATLAB pca documentation, 2018* (MathWorks). AL simulations were run in Python (version 3.6) (*Rossum and Drake, 2011*), and model outputs were analyzed using Jupyter notebooks (*Kluyver et al., 2016*) and Python scripts. We performed a power analysis prior to the study to determine that recording calcium activity in 20–40 flies would be sufficient to identify moderate calcium–behavior correlations. Sample sizes for expansion microscopy were smaller, as the experimental procedure was more involved – therefore, we did not conduct a formal statistical analysis. Linear models were fit using the fitlm MATLAB function (https://www.mathworks.com/help/stats/fitlm.html); coefficients and p values of models between measured preferences and predicted preferences are listed in *Table 1*. 95% CIs around model regression lines were estimated as ±2 SDs of the value of the regression line at each x-position across 2000 bootstrap replicates (resampling flies). Boxplots depict the median value (points), interquartile range (boxes), and range of the data (whiskers).

## Acknowledgements

We thank Asa Barth-Maron and Rachel Wilson for discussions helpful to the AL modeling, and Katrin Vogt for help revising the manuscript. Ed Soucy and Brett Graham of the Center for Brain Science Neuroengineering Core helped maintain the olfactometer and microscope. DL was supported by the NSF-Simons Center for Mathematical and Statistical Analysis of Biology at Harvard, award number #1764269 and the Harvard Quantitative Biology Initiative. BLdB was supported by a Klingenstein-Simons Fellowship Award, a Smith Family Odyssey Award, a Harvard/MIT Basic Neuroscience Grant, National Science Foundation grant no. IOS-1557913, and NIH/NINDS grant no. 1R01NS121874-01. EB was supported by a Harvard/MIT Basic Neuroscience Grant, Lisa Yang, John Doerr, and NIH grant no. 1R01EB024261.

# Additional information

## Competing interests

Ruixuan Gao: co-inventor on multiple patents related to expansion microscopy. Edward S Boyden: co-founder of a company that aims to commercialize expansion microscopy for medical purposes; co-inventor on multiple patents related to expansion microscopy. The other authors declare that no competing interests exist.

## Funding

| Funder | Grant reference number | Author |
| --- | --- | --- |
| Klingenstein-Simons Fellowship Award | | Benjamin L de Bivort |
| Smith Family Foundation | Odyssey Award | Benjamin L de Bivort |
| National Science Foundation | IOS-1557913 | Benjamin L de Bivort |
| National Institute of Neurological Disorders and Stroke | 1R01NS121874-01 | Benjamin L de Bivort |
| National Institutes of Health | 1R01EB024261 | Edward S Boyden |
| NSF-Simons Center for Mathematical and Statistical Analysis of Biology at Harvard | 1764269 | Danylo O Lavrentovich |
| Harvard/MIT Basic Neuroscience Grant | | Edward S Boyden Benjamin L de Bivort |

The funders had no role in study design, data collection and interpretation, or the decision to submit the work for publication.

## Author contributions

Matthew A Churgin, Danylo O Lavrentovich, Conceptualization, Data curation, Software, Formal analysis, Investigation, Visualization, Methodology, Writing – original draft, Writing – review and editing; Matthew A-Y Smith, Investigation, Methodology, Writing – review and editing; Ruixuan Gao, Resources, Data curation, Supervision, Methodology, Writing – review and editing; Edward S Boyden, Supervision, Funding acquisition, Methodology, Writing – review and editing; Benjamin L de Bivort, Conceptualization, Software, Formal analysis, Supervision, Funding acquisition, Visualization, Methodology, Writing – original draft, Project administration, Writing – review and editing

## Author ORCIDs

Matthew A Churgin (ID) https://orcid.org/0000-0003-2299-0124
Danylo O Lavrentovich (ID) https://orcid.org/0000-0002-8432-9596
Matthew A-Y Smith (ID) https://orcid.org/0000-0003-0913-1392
Benjamin L de Bivort (ID) https://orcid.org/0000-0001-6165-7696

Joint public review: https://doi.org/10.7554/eLife.90511.4.sa1
Author response https://doi.org/10.7554/eLife.90511.4.sa2

# Additional files

## Supplementary files

MDAR checklist

## Data availability

All raw data, totaling 600 GB, are available via hard drive from the authors. A smaller (7 GB) repository with partially processed data files and MATLAB/Python scripts sufficient to generate figures and results is available at Zenodo (https://doi.org/10.5281/zenodo.14252278).

The following dataset was generated:

| Author(s) | Year | Dataset title | Dataset URL | Database and Identifier |
|---|---|---|---|---|
| Churgin M, Lavrentovich D, Smith M, Gao R, Boyden E, de Bivort BL | 2024 | Data for: A neural correlate of individual odor preference in Drosophila | https://doi.org/10.5281/zenodo.14252278 | Zenodo, 10.5281/zenodo.14252278 |

The following previously published dataset was used:

| Author(s) | Year | Dataset title | Dataset URL | Database and Identifier |
|---|---|---|---|---|
| Münch D, Galizia CG, Strauch M, Nissler A, Ma S | 2016 | DoOR 2.0 - Comprehensive Mapping of *Drosophila melanogaster* Odorant Responses | https://doi.org/10.5281/zenodo.46554 | Zenodo, 10.5281/zenodo.46554 |

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
