## [Editor Report · eLife Assessment]

What makes one member of the species behave differently from another? This is a core problem in behavioral neuroscience. This **valuable** study seeks an answer for the specific case of the fruit fly expressing preferences for one odor over another. By a combination of behavioral measurements, neurophysiology, and network modeling, the authors find **solid** evidence for at least one locus of individuality in the peripheral olfactory system.

---

## [Referee Report · Joint public review]

Summary:

The authors aimed to identify the neural sources of behavioral variation in fruit flies deciding between odor and air, or between two odors.

Strengths:

- The question is of fundamental importance.

- The behavioral studies are automated, and high-throughput.

- The data analyses are sophisticated and appropriate.

- The paper is clear and well-written aside from some initially strong wording.

- The figures beautifully illustrate their results.

- The modeling efforts mechanistically ground observed data correlations.

Weaknesses:

-The correlations between behavioral variations and neural activity/synapse morphology are statistically significant but relatively weak.

---

## [Author Response]

The following is the authors’ response to the previous reviews.

**Joint Public Review:**
Summary:The authors aimed to identify the neural sources of behavioral variation in fruit flies deciding between odor and air, or between two odors.Strengths:- The question is of fundamental importance.- The behavioral studies are automated, and high-throughput.- The data analyses are sophisticated and appropriate.- The paper is clear and well-written aside from some initially strong wording.- The figures beautifully illustrate their results.- The modeling efforts mechanistically ground observed data correlations.Weaknesses:- The correlations between behavioral variations and neural activity/synapse morphology are relatively weak, and sometimes overstated in the wording that describes them.

We sincerely thank the reviewers for these evaluations.

**Recommendations for the authors:**
Line 56: "We hypothesize that as sensory cues are encoded and transformed to produce motor outputs, their representation in the nervous system becomes increasingly idiosyncratic and predictive of individual behavioral responses". This seems obvious a priori. The sensory stimuli are the same, but the motor responses are different. Along the way there has to be a progression from same to different. Is there an alternative hypothesis? If so, perhaps state the alternative.

We added text to the first paragraph of the introduction (lines 58-60) laying out an alternative hypothesis that individuality emerges through biomechanical differences and environmental interactions, and we have altered our motivating question to assess *whether* circuit elements in which activity is predictive of individual behavior exist, and if so, where (lines 60-62).

Line 157: typo "remaining"

We changed “remaining” to “remain” (line 160).

Line 163: why report r sometimes and R^2 other times? Better to use R^2 throughout.

We changed all instances of r to R^2^, notably when reporting combined train/test statistics for calcium - behavior models (line 162). We also reframed the outputs (medians + 90% confidence intervals) of the supplemental analysis inferring the strength of the latent calcium-behavior relationship to be in terms of R^2^ (lines 166, 173-175, 241, 252; modified text in *Inference of correlation between latent calcium and behavior states* in Materials and Methods; adjusted figure and caption for Figure 1 – figure supplement 9).

Line 182: "odorant". Should be "odorant receptors"?

We respectfully disagree – our ORN and PN calcium data are responses to odorants in 5 glomerulus/odorant receptor types. When we group PCA loadings by glomerulus for both ORN and PN calcium, the consistency within groups is much stronger than when we group the loadings by odorant (Figure 1 – figure supplement 8). Additionally, “odorant receptor organization” would mean the same thing as “glomerular organization,” since all ORNs expressing the same odorant receptor project to a single glomerulus.

Line 331: "harbor". Maybe more modestly "contribute to"?

We changed “harbor” to “contribute to” (line 334) and added additional moderating language that the difference in DC2 and DM2 activations in PNs explains a large portion of the individuality signal (lines 337-339).

Line 403: typo "is"

We retained “is” as the corresponding verb for “the net effect,” but we adjusted the position of the reference to Gomez-Marin and Ghazanfar, 2019 for more clarity (lines 406-408).